# Hepatitis C Virus Infection at Primary Healthcare Level in Abha City, Southwestern Saudi Arabia: Is Type 2 Diabetes Mellitus an Associated Factor?

**DOI:** 10.3390/ijerph15112513

**Published:** 2018-11-09

**Authors:** Suliman M. Al Humayed, Ahmed A. Mahfouz, Nabil J. Awadalla, Abdullah A. Alsabaani

**Affiliations:** 1Department of Internal Medicine, College of Medicine, King Khalid University, P.O. Box 641, Abha 61421, Saudi Arabia; s_humayed@yahoo.com; 2Department of Family and Community Medicine, College of Medicine, King Khalid University, P.O. Box 641, Abha 61421, Saudi Arabia; njgirgis@yahoo.co.uk (N.J.A.); dr.alsabaani@hotmail.com (A.A.A.); 3Department of Epidemiology, High Institute of Public Health, Alexandria University, Alexandria 21511, Egypt; 4Department of Community Medicine, College of Medicine Mansoura University, Mansoura 35516, Egypt

**Keywords:** hepatitis C virus infection, type 2 diabetes mellitus, primary healthcare centers, Saudi Arabia

## Abstract

Background: There is an increasing concern about the relation between hepatitis C virus infection (HCV) and type 2 diabetes mellitus (T2DM). The present study aims to determine the prevalence of HCV infection among T2DM patients and non-diabetic patients attending primary healthcare centers (PHCCs) in Abha city, southwestern Saudi Arabia, and to explore the possible association between T2DM and HCV infection. Methods: A cross-sectional study targeting a random sample of T2DM and non-diabetic patients attending PHCCs in Abha City was conducted. Patients were interviewed using a structured questionnaire and screened for HCV infection using fourth-generation ELISA kits. All positive cases were confirmed by qualitative RT-PCR immune assay. Results: The study revealed an overall seroprevalence of HCV infection of 5% (95% CI: 2.9–7.9%). Among T2DM and non-diabetics, a seroprevalence of 8.0% and 2.0% was found, respectively. Using multivariable regression analysis, the only significant associated factor for HCV infection was T2DM (aOR = 4.185, 95% CI: 1.074–16.305). Conclusions: There is strong positive association between T2DM and HCV infection. Yet, the direction of relationship is difficult to establish. Patients with T2DM have higher prevalence of HCV infection than non-diabetic group. It is highly recommended for primary health care providers to screen for HCV infection among T2DM patients and to increase the level of HCV awareness among them.

## 1. Introduction

Type 2 diabetes (T2DM) is a common endocrine disorder that involves multi-factorial mechanisms. These mechanisms include resistance to the action of insulin, increased hepatic glucose production, and a defect in insulin secretion, all of which contribute to the development of hyperglycemia and diabetes [1]. In a cross-sectional study enrolling more than 6000 patients in Riyadh, the prevalence of diabetes was found to be 30% [2]. This prevalence is higher than that reported from other Gulf countries [3,4]. The majority of diabetic patients in Saudi Arabia are managed in primary health care centers. Clearly, the health and the financial burdens of the disease in Saudi Arabia are immense.

HCV infection is another burden on both health and finances. It is estimated globally that three to four million are newly infected each year; 170 million are chronically infected and are at risk of developing chronic liver disease, including cirrhosis and hepatocellular carcinoma [5].

There are increasing reports indicating a possible association between T2DM and HCV infection [6]. In Saudi Arabia, most of the studies on HCV seroprevalence have been performed on certain population groups. A figure of 1.43% among Saudi blood donors was reported in the Aseer region, Southwestern Saudi Arabia [7]. Other studies in Saudi Arabia reported HCV infection among hemodialysis patients [8] and among persons attending premarital screening programs [9]. Yet, data from patients attending primary health care centers are scarce.

Abha is the capital of the Aseer region, Southwestern Saudi Arabia. The aim of the present work was to study the prevalence of HCV infection among patients attending PHCCs in Abha City and to explore the possible association between T2DM and HCV infection.

## 2. Materials and Methods

### 2.1. Study Design and Settings

The study was a cross-sectional study targeting patients attending PHCCs in Abha City. The main inclusion criteria were adult patients (18 years and above).

### 2.2. Sample Size and Sampling

Using the WHO manual for Sample Size Determination in Health Studies [10,11], at a 95% confidence interval with an estimate of the anticipated prevalence of 1.65% [11], and with an absolute precision of 2%, the minimal sample size required for the study was calculated to be 165 patients. To avoid loss of cases, a total sample of 300 patients was planned to be included in the study. Patients were selected from the 7 Abha City PHCCs using a systematic random sampling method to ensure a high degree of randomization.

### 2.3. Questionnaire Interview

Participants were interviewed. The interview questionnaire incorporated socio-demographic data and history of past medical history and relevant risk factors. History of blood transfusion was specifically inquired about. Data regarding history of surgical operations and tooth extraction were also collected. Weight and standing height were measured. Body mass index (BMI) was calculated as weight(kg)/height (m^2^).

### 2.4. Blood Sampling

Blood samples were withdrawn. Samples were then centrifuged and stored at −20 °C until they were transported in ice boxes to the Virus Laboratory of Abha College of Medicine, King Khalid University.

### 2.5. Serologic Testing for HCV Antibodies

Diagnosis of HCV infection was performed using fourth-generation ELISA. Kits were obtained from DIA-PRO Diagnostic Bioprobes Sr1 Via G. Carducci, Milano, Italy. The assay detects IgG class antibodies against the viral structural protein c-22-3 derived from the genomic core region and three non-structural proteins (c-33c, c100-3, and 5-1-1) derived from the NS3 and NS4 regions of the viral genome. The company’s protocols were strictly applied in testing and interpreting the results. All positive and equivocal HCV serology results were further confirmed by utilizing a qualitative confirmatory RT-PCR.

### 2.6. Statistical Analysis

Arithmetic mean, standard deviation, and proportions (with 95% confidence intervals) were used to present the data. Crude odds ratios (cOR) and 95% confidence intervals (95% CI) were calculated for univariate analysis. The logistic regression model was used to evaluate factors associated with HCV. Adjusted odds ratios (aOR) and their 95% CI were calculated. Variables included in the model were gender, age, education, BMI, having T2DM, history of blood transfusion, tooth extraction, and surgical operations.

### 2.7. Ethical Approval

Approvals were taken from the Research Ethics Committee, College of Medicine, King Khalid University (HA-O8-B-017, dated July 2016). Written informed consents were obtained from all patients enrolled in the study. Researchers also got the permission of the Aseer Directorate of Health Affairs.

## 3. Results

The present study included 300 patients attending the urban primary health care centers in Abha City. Table 1 summarizes characteristics of the study sample. The age ranged from 25 to 100 years, with an average of 55.9 ± 13.4 years and a median of 55 years. The sample included 187 males (62.3%) and 113 females (37.7%). The mean age of males (56.8 ± 13.7 years) was not statistically different (t = 1.53, *p* = 0.127) from females (54.8 ± 13.0 years). The majority of the sample were married (80%, 240), educated (205, 68.3%), and non-smokers (223, 74.3%). The present study included 150 type 2 diabetes mellitus (T2DM) patients from diabetes clinics and 150 non-diabetics.

Overall, fifteen patients were found to be positive for HCV infection. The study revealed an overall seroprevalence of HCV infection of 5% (95% CI: 2.9–7.9%). The prevalence of HCV among T2DM patients amounted to 8.0%. On the other hand, the prevalence of HCV among non-diabetics amounted to 2.0%. The study showed that T2DM patients had significantly four times the risk to have HCV infection (cOR: 4.261, 95% CI: 1.177–15.422).

Table 2 shows that in a multivariable regression analysis for the factors associated with HCV among the study sample, only T2DM patients had significantly four times the risk (aOR = 4.185, 95% CI: 1.074–16.305) to have HCV compared to non-diabetics. On the other hand, age, gender, level of education, BMI, history of blood transfusion, tooth extraction, and surgery were found to be of no statistical significance.

## 4. Discussion

The present study reported an overall prevalence of HCV infection among primary health care attendants in Abha City, Southwestern Saudi Arabia, of 5%. This figure seems to be higher than the estimated global prevalence of 2–3% [12] and the estimated prevalence in Saudi Arabia of 1.0–3.0% [13]. The reported higher prevalence of HCV in the current study could be explained firstly by the unique characters of patients attending chronic disease clinics at primary healthcare centers and included in the study, as 50% of them were diabetic. The result of the present study showed that the seropositive prevalence of HCV among diabetic patients was 8.0%. Furthermore, 45−85% of the infected adult population lives unrecognized and are discovered accidently when receiving medical care for non-hepatic complaints [14].

The prevalence rate of HCV infection among T2DM patients in the current study (8.0%) is much higher than that reported in Greece (1.65%) [15], Dammam, Saudi Arabia (1.9%) [16], and France (3%) [17]. On the other hand, it is much lower than the rates reported from Pakistan (36%) [18] and Nigeria (11%) [19].

The present study revealed a prevalence rate of HCV infection among non-diabetic patients of 2.0%. This figure is comparable to the estimated prevalence rate among general population in Southwestern Saudi Arabia of 2.2% (95% CI: 1.9–2.5%) [20].

The results of the multivariable logistic regression revealed that T2DM patients were 4 times more likely to have HCV compared to non-diabetic patients. This suggests a strong positive significant association between T2DM and HCV infection. Recently published review articles [6,21] revealed similar findings. Studies in Pakistan [22] and in Dammam, Saudi Arabia [16], confirmed this finding. The relation between HCV infection and diabetes may be bi-directional [23]. It is difficult to establish the direction of the relationship between HCV infection and T2DM by using the cross-sectional design of our study. The obvious biological link between HCV infection and T2DM has not been confirmed [6]. The higher possibility of T2DM patients to have HCV infection could possibly be the outcome of the biological effect of chronic HCV infection on the pathogenesis of diabetes [24]. On the other hand, presence of T2DM was proposed to enhance HCV infection through the immunocompromised state of diabetics [12]. Similarly, the presence of hyperinsulinemia observed in diabetic patients [25] and frequent percutaneous procedures during insulin therapy and glycemic assessment [6] may play a role. The association of chronic HCV infection, without cirrhosis, with T2DM, has been shown by other studies, but this is debated [26]. The present study was not addressing this point. Similarly, a recent meta-analysis study showed that HCV infection is associated with an increased risk of T2DM separately from the severity of the associated liver disease [27].

There are some limitations to the present study. First, a disadvantage of the cross-sectional study design is its inherent inability to establish the direction of the relationship between HCV infection and T2DM. Second, the study has a relatively small sample size, which may be unable to explore in depth the other factors associated with HCV infection among diabetic patients. The generalizability of our results on the national basis cannot be established.

## 5. Conclusions

There is a strong positive association between T2DM and HCV infection. Yet, the direction of this relationship is difficult to establish in the current study because of the cross-sectional design. Patients with T2DM have a higher prevalence of HCV infection than the non-diabetic group.

The current study demonstrates the need to conduct longitudinal studies to establish the temporality of the relationship. Additionally, studies with larger samples are recommended for the in-depth exploration of factors associated with higher risk of HCV infection among diabetic patients. It is highly recommended for primary health care providers to screen for HCV infection among T2DM patients and to increase the level of HCV awareness among them.

## Figures and Tables

**Table 1 ijerph-15-02513-t001:** Personal characteristics of the study sample of patients attending primary health care centers (*n* = 300).

Characteristics		NO. (%)
**Age (years)**	<40 years	36 (12.0)
	40–60	145 (48.3)
	≥60	119 (39.7)
**Gender**	Male	187 (62.3)
	Female	113 (37.7)
**Marital status**	Single	60 (20.0)
	Married	240 (80.0)
**Education**	Illiterate	95 (31.7)
	Primary/Intermediate	91 (30.3)
	Secondary	54 (18.0)
	University	60 (20.0)
**Smoking status**	Current Smoker	44 (14.7)
	Ex-smoker	33 (11.0)
	Non-smoker	223 (74.3)

**Table 2 ijerph-15-02513-t002:** Multivariable analysis for factors associated with viral hepatitis C seropositivity among patients attending primary health care centers (*n* = 300).

Factors	aOR (95% CI)
Age (years)	1.030 (0.984–1.077)
Female vs. Male	1.169 (0.336–4.061)
BMI (kg/m^2^)	1.028 (0.936–1.130)
T2DM vs. Non T2DM	**4.185 (1.074–16.305)**
Positive history of blood transfusion	0.653 (0.074- 5.760)
Positive history of Tooth extraction	4.438 (0.525–37.548)
Positive history of surgery	0.457 (0.147–1.418)
Secondary or higher education vs. less than secondary	0.835 (0.190–3.666)

T2DM = Type 2 Diabetes Mellitus, aOR = adjusted Odds Ratio, 95% CI = 95% Confidence Interval, Bold OR = Significant.

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
