# Peer review of "Hepatitis C Virus Infection at Primary Healthcare Level in Abha City, Southwestern Saudi Arabia: Is Type 2 Diabetes Mellitus an Associated Factor?"

_ijerph, 2018, doi:10.3390/ijerph15112513_

Round 1
Reviewer 1 Report
The research is of great importance to health care providers and workers who are responsible for the welfare of type 2 diabetic patients. The methodology was acceptable for the type of testing done. To minimize the potential for false negatives, i.e. ELISA testing before antibody development, duplicate testing after approximately three months might have improved the soundness of the research. That being said, it was an important primary study.
There are a few minor corrections needed:
Page 4. Line 128. After "adult" the words, "populationlive" unrecognized and "dicovered" need to be corrected or the sentence be re-phrased.
Page 4. Line 132. The phrase, "On the hand" perhaps should be restated as " On the other hand".
Page 6. Line 210. Insert space between "on" and "nosocomial".
Page 6. Line 210. After "devices" insert space after "on" and before "nosocomial".
Author Response
RESPONSE TO COMMENTS OF REVIEWER ONE
IJERPH 3788027
· The authors are grateful to the reviewer for his fruitful comments.
· Changes were made in the article using "track changes".
· Spell check and language style were revised.
· Page 4. Line 128. After "adult" the words, "populationlive" unrecognized and "dicovered" need to be corrected or the sentence be re-phrased. Corrected
· Page 4. Line 132. The phrase, "On the hand" perhaps should be restated as " On the other hand". Corrected
· Page 6. Line 210. Insert space between "on" and "nosocomial". Corrected
· Page 6. Line 210. After "devices" insert space after "on" and before "nosocomial". Corrected

Reviewer 2 Report
The study aims to evaluate the possible association between T2DM and HCV infection.The study revealed among T2DM and non-diabetics a seroprevalence of 8.0% and 2.0% respectively. Using multivariate regression analysis, the only significant associated factor for HCV infection was T2DM.
The study might be interesting, however some points need to be addressed.
1) The association of cirrhosis of whatever origin with T2DM, is well known. The association of chronic HCV infection, without cirrhosis, with T2DM, has been shown, by other studies, and however is under debate. So the Authors could show if the association is present or not in patients with chronic HCV infection, without cirrhosis. If Authors cannot show results this point needs to be at least discusssed. See for example: Diabetes Care 28(10), pp. 2548-2550; World Journal of Hepatology 7(3), pp. 327-343.
2) A recent meta-analysis has been published about the association of HCV infection, with T2DM. Please discuss this point. See: Rev Endocr Metab Disord. 2018 Jan 11. doi: 10.1007/s11154-017-9440-1.
3) English language needs to be revised.
Author Response
RESPONSE TO COMMENTS OF REVIEWER ONE
IJERPH 3788027
· The authors are grateful to the reviewer for his fruitful comments.
· Changes were made in the article using "track changes".
· The association of chronic HCV infection, without cirrhosis, with T2DM, has been shown, by other studies, and however is under debate. The present study was not addressing this point. This statement was added at the end of the discussion section.
· The meta analysis study results was added at the end of the discussion.

Round 2
Reviewer 2 Report
The paper is improved after the revision.